# Key factors for connecting silver-based icosahedral superatoms by vertex sharing

Sayuri Miyajima[1], Sakiat Hossain [2 ✉], Ayaka Ikeda[1], Taiga Kosaka[1], Tokuhisa Kawawaki[1,2], Yoshiki Niihori[2], Takeshi Iwasa [3,4 ✉], Tetsuya Taketsugu[3,4] & Yuichi Negishi [1,2 ✉]

Metal nanoclusters composed of noble elements such as gold (Au) or silver (Ag) are regarded as superatoms. In recent years, the understanding of the materials composed of superatoms, which are often called superatomic molecules, has gradually progressed for Au-based materials. However, there is still little information on Ag-based superatomic molecules. In the present study, we synthesise two di-superatomic molecules with Ag as the main constituent element and reveal the three essential conditions for the formation and isolation of a superatomic molecule comprising two $Ag_{13-x}M_x$ structures (M = Ag or other metal; $x$ = number of M) connected by vertex sharing. The effects of the central atom and the type of bridging halogen on the electronic structure of the resulting superatomic molecule are also clarified in detail. These findings are expected to provide clear design guidelines for the creation of superatomic molecules with various properties and functions.

[1] Department of Applied Chemistry, Faculty of Science, Tokyo University of Science, Kagurazaka, Shinjuku—ku, Tokyo 162—8601, Japan. [2] Research Institute for Science & Technology, Tokyo University of Science, Kagurazaka, Shinjuku—ku, Tokyo 162—8601, Japan. [3] Department of Chemistry, Faculty of Science, Hokkaido University, Sapporo, Hokkaido 060—0810, Japan. [4] WPI-ICReDD, Hokkaido University, Sapporo, Hokkaido 060—0810, Japan. ✉email: sakiathossain@rs.tus.ac.jp; tiwasa@sci.hokudai.ac.jp; negishi@rs.tus.ac.jp

Metal nanoclusters (NCs)[1–14] composed of noble metal elements such as gold (Au) and silver (Ag) are stabilised when the total number of valence electrons satisfies the closed-shell electronic structure, as in conventional atoms[15,16]. Such metal NCs are regarded as superatoms (artificial atoms). If superatoms are used to assemble materials, it might be possible to create materials with physicochemical properties and functions that are different from those of conventional materials[17]. Regarding such materials composed of superatoms (often called superatomic molecules[18,19]), since the 1980s, there have been many reports of Au-based superatomic molecules, which Teo and Zhang called clusters of clusters[20]. Subsequent work by groups such as Tsukuda[21], Nobusada[22], Jin[23] and Zhu[24] has gradually improved our understanding of the types of superatomic molecules that can be produced and the electronic structures that can be created[25]. Ag NCs have multiple properties and functions that are superior to those of Au NCs, including photoluminescence (PL) with high quantum yield[26] and selective catalytic activity for carbon dioxide reduction[27]. However, there are only a limited number of reports, including the report[28] by the authors, on Ag-based superatomic molecules[29–32]. To construct substances using superatomic molecules and create new materials, it is essential to gain a deeper understanding of the types of superatomic molecules that can be produced and the electronic structures that can be created, even for Ag-based superatomic molecules.

In the present study, we focus on Ag-based 13-atom NCs ($Ag_{13-x}M_x$; M = Ag or other metal; $x$ = number of M) as superatoms, and aim to elucidate the key factors in the formation of di-superatomic molecules by vertex sharing[33] and the electronic structure of the obtained di-superatomic molecules. Platinum (Pt) or palladium (Pd) was used as the element that substitutes part of the Ag, and chloride (Cl) or bromide (Br) was used as the bridging ligand to support the connection of the two 13-atom NCs. To achieve our purpose, in addition to two previously reported di-superatomic molecules ($[Ag_{23}Pt_2(PPh_3)_{10}Cl_7]^0$ (**1**); Fig. 1a; PPh₃ = triphenylphosphine)[31] and ($[Ag_{23}Pd_2(PPh_3)_{10}Cl_7]^0$ (**2**); Fig. 1b)[28], we synthesised two new superatomic molecules with Br as the bridging ligand ($[Ag_{23}Pt_2(PPh_3)_{10}Br_7]^0$ (**3**) and $[Ag_{23}Pd_2(PPh_3)_{10}Br_7]^0$ (**4**); Table 1). We investigated their geometric/electronic structures and their stabilities with regard to

degradation in solution. Consequently, we confirmed that **3** and **4** both have a geometric/electronic structure that qualifies them as superatomic molecules. Regarding the electronic structure, we further observed that (1) there is a peak attributable to the metal core at approximately 600 nm in the optical absorption spectra of all the superatomic molecules; (2) such peaks shift to longer wavelengths when M is changed from Pt to Pd; (3) all **1−4** exhibit PL in visible-to-near infra-red (NIR) region; and (4) PL peaks shift to longer wavelengths when M is changed from Pt to Pd. With respect to the stability of the superatomic molecule described by $[Ag_{23}M_2(PPh_3)_{10}X_7]^z$ (M = Ag, Pd, or Pt; X = Cl or Br; z = 2+ or 0), we found that the stability decreases in the order **1 > 3 > 2 > 4** (which can be synthesised) > $[Ag_{25}(PPh_3)_{10}X_7]^{2+}$ (X = Cl or Br; which are not so stable in solution). Based on these results and reports on the related superatomic molecules, we concluded that

**Table 1 NC number, chemical composition, number of bridging halogens, number of total valence electrons, and references to literature on Ag-based di-superatomic molecules described in the present paper.**

| NC | Chemical composition[a] | $N_{bx}$[b] | $N_{te}$[c] | Ref. |
|---|---|---|---|---|
| 1 | $[Ag_{23}Pt_2(PPh_3)_{10}Cl_7]^0$ | 5 | 16 | 31 |
| 2 | $[Ag_{23}Pd_2(PPh_3)_{10}Cl_7]^0$ | 5 | 16 | 28 |
| 3 | $[Ag_{23}Pt_2(PPh_3)_{10}Br_7]^0$ | 5 | 16 | p.w.[d] |
| 4 | $[Ag_{23}Pd_2(PPh_3)_{10}Br_7]^0$ | 5 | 16 | p.w.[d] |
| 5 | $[Au_{23}Pd_2(PPh_3)_{10}Br_7]^0$ | 5 | 16 | 40 |
| 6 | $[Au_{13}Ag_{12}(P(p\text{-}Tol)_3)_{10}Cl_7](SbF_6)_2$ | 5 | 16 | 41 |
| 7 | $[Au_{13}Ag_{12}(PPh_3)_{10}Cl_8](SbF_6)$ | 6 | 16 | 42 |
| 8 | $[Au_{13}Ag_{12}(PPh_3)_{10}Br_8](SbF_6)$ | 6 | 16 | 43 |
| 9 | $[Au_{13}Ag_{12}(P(p\text{-}Tol)_3)_{10}Br_8](PF_6)$ | 6 | 16 | 44 |
| 10 | $[Au_{13}Ag_{12}(PPh_3)_{10}Br_8]Br$ | 6 | 16 | 45 |
| 11 | $[Au_{12}Ag_{13}((PMePh_2)_{10}Br_9]^0$ | 7 | 16 | 46 |
| 12 | $[Au_{11}Ag_{12}Pt_2(PPh_3)_{10}Cl_7]^0$ | 5 | 16 | 49 |
| 13 | $[Au_{10}Ag_{13}Pt_2(PPh_3)_{10}Cl_7]^0$ | 5 | 16 | 50 |

[a]$SbF_6^-$ hexafluoroantimonate, $PF_6^-$ hexafluorophosphonate, $P(p\text{-}Tol)_3$ tri($p$-tolyl)phosphine.
[b]Number of bridging halogens.
[c]Number of total valence electrons.
[d]The present work.

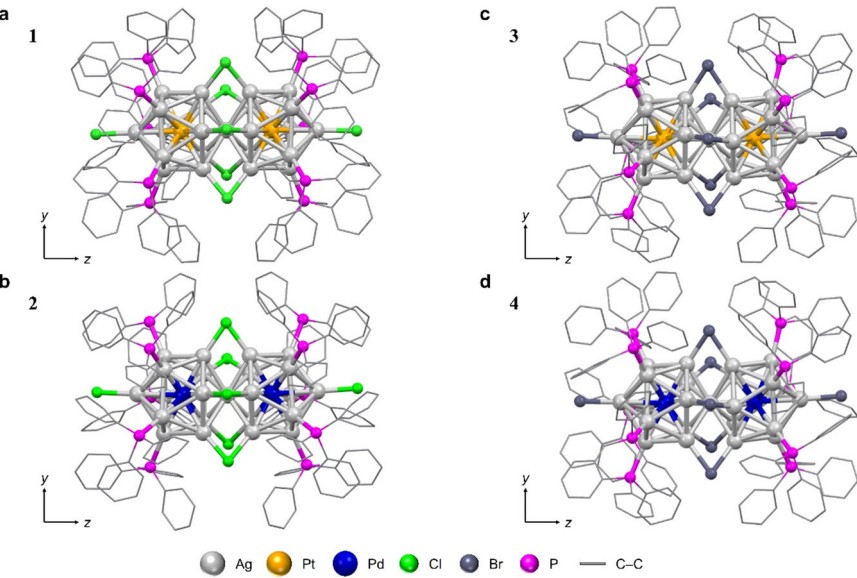

Ag    Pt    Pd    Cl    Br    P    —— C–C

**Fig. 1 Comparison of the geometric structures. a 1. b 2. c 3. d 4.** The geometric structure of **1** and **2** are reproduced from ref. [28,31], respectively (grey = Ag; orange = Pt; blue = Pd; green = Cl; dark grey = Br; magenta = P). The positions of the Pd atoms are the predicted positions based on DFT calculations.

the following three conditions are essential for the formation and isolation of a superatomic molecule consisting of two $Ag_{13-x}M_x$ structures (M = Ag or other metal) connected by vertex sharing ($[Ag_{25-x}M_x(PR_3)_{10}X_y]^z$; $PR_3$ = phosphine; $y$ = number of X): (1) a halogen ligand of a size that can maintain a moderate distance between two $Ag_{13-x}M_x$ structures is used as the bridging ligand; (2) an icosahedral core, which is stronger than $Ag_{13}$, is formed by heteroatom substitution; and (3) $[Ag_{25-x}M_x(PR_3)_{10}X_y]^z$ comprises substituted heteroatoms and bridging halogens such that the total number of valence electrons is 16 when they are cationic or neutral.

## Results and discussion

**Synthesis and geometric structure.** An NC mixture containing **3** was first prepared by adding sodium borohydride ($NaBH_4$) methanol solution to a methanol solution containing silver nitrate ($AgNO_3$), $PtBr_2$, $PPh_3$, and NaBr in the dark. The by-products were then removed by washing with the solvent, and the product was crystallised to obtain high-purity **3** (Fig. S1a)[28]. Electrospray ionisation-mass spectrometry (ESI-MS) of the product showed that **3** has a chemical composition of $[Ag_{23}Pt_2(PPh_3)_{10}Br_7]^0$ (Fig. S2). X-ray photoelectron spectroscopy (XPS; Fig. S3) confirmed the presence of Pt in **3**. We also obtained **4** as single crystals using a process similar to that used to synthesise **3**, except that $PdBr_2$ was used instead of $PtBr_2$ (Fig. S1b). X-ray photoelectron spectroscopy (XPS; Ag : Pd = 23 : 1.5; Fig. S4) confirmed the presence of Pd in **4**.

Figure 1c shows the geometric structure of **3** determined by single crystal X-ray diffraction (SC-XRD) analysis (Fig. S5, Table S1 and Supplementary Data 1, 2). We found that **3** has a geometric structure in which two icosahedral $Ag_{12}Pt$ molecules are connected by vertex sharing. Pt was located at the central position in each icosahedral $Ag_{12}Pt$ molecule, as often seen in the literatures[34,35]. This structure is similar to the geometric structure of **1**, as previously reported (Fig. 1a)[31]. The SC-XRD analysis of **3** did not confirm the presence of counter ions (Fig. S5), again supporting the interpretation that **3** was isolated as a neutral NC ($[Ag_{23}Pt_2(PPh_3)_{10}Br_7]^0$). These results demonstrate that both **1** and **3** have 16 valence electrons (Table 1)[16]. Therefore, the two $Ag_{12}Pt$ structures in **3** are described as $[Ag_{12}Pt]^{4+}$, indicating that both have a closed-shell electronic structure that satisfies the $1S^2 1P^6$ superatom orbital (Fig. S6)[15,19]. We concluded from these results that **3** is a NC that can be regarded as a di-superatomic molecule, similar to **1**.

Figure 1d shows the geometric structure of **4** (Fig. S7, Table S1 and Supplementary Data 3, 4). As you can see, **4** has a geometric structure in which two icosahedral $Ag_{12}Pd$ structures are connected by vertex sharing, which is similar to the geometric structure of the previously reported **2** (Fig. 1b)[28]. In addition, **4** was also isolated as a neutral molecule (Fig. S7), indicating that each $Ag_{12}Pd$ structure in **4** has a closed-shell electronic structure that satisfies $1S^2 1P^6$ (Fig. S6)[15,19]. We concluded from these results that **4** is also a NC that can be considered a di-superatomic molecule.

Unfortunately, it is difficult to determine the Pd position for **4** by SC-XRD alone because Pd ($_{46}$Pd) and Ag ($_{47}$Ag) have a similar number of electrons. However, Pd (1.920 J m$^{-2}$ for Pd(111))[36] has a higher surface energy than Ag (1.172 J m$^{-2}$ for Ag(111))[36], and Pd is generally located in the centre of the icosahedral structure in $Ag_{12}Pd$[28,34,35,37,38]. We performed density functional theory (DFT) calculations for $[Ag_{23}Pd_2(PPh_3)_{10}Br_7]^0$ with different Pd positions using the Perdew–Burke–Ernzerhof (PBE) functional to confirm that Pd is located at the centre of the two icosahedral structures in **4** as in **2**. The results showed that $[Ag_{23}Pd_2(PPh_3)_{10}Br_7]^0$ is stable for Pd positions in the order of the icosahedral centre (i) > the icosahedral surface (ii) > the

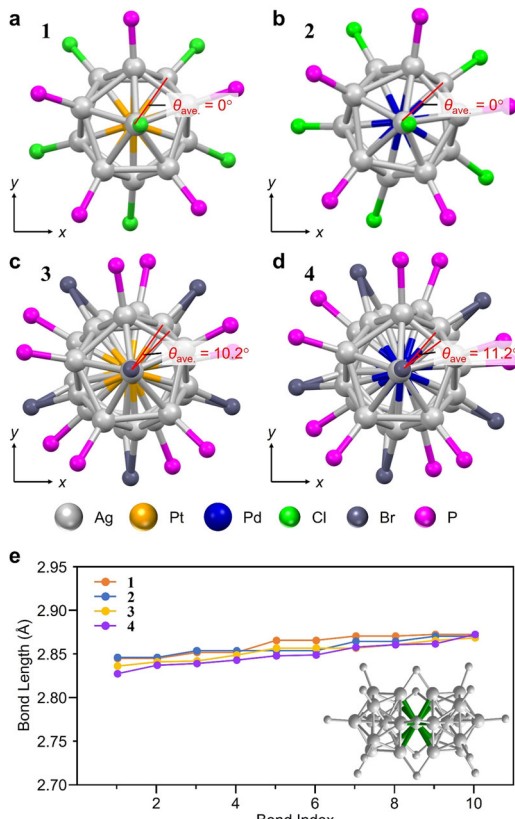

**Fig. 2 Structural analysis of the twist between the two $Ag_{12}M$ structures (M = Pt or Pd). a–d** View from the long-axis direction for **1**, **2**, **3** and **4**, respectively, showing the twist between the two $Ag_{12}M$ structures (M = Pt or Pd) in the cores of **3** and **4** (grey = Ag; orange = Pt; blue = Pd; green = Cl; dark grey = Br; magenta = P). The geometric structure of **1** and **2** are reproduced from ref. [28,31], respectively. In **a–d**, $\theta_{ave}$ indicates the average dihedral angle between the two $Ag_{12}M$ structures. **e** Comparison of the Ag−Ag bond length between the joining Ag and the neighbouring Ag (green line), showing that the bond lengths are quite similar in **1**−**4**.

shared vertex (iii) (Fig. S8). Based on these results, we concluded that the two Pd atoms are located in the centre of the icosahedral structure in **4** (Fig. 1d), as in **2**.

In this way, overall **1**−**4** have similar geometric structures. However, a detailed look at their geometric structures revealed some differences between **1** and **2**, which use Cl as the bridging halogen, as well as between **3** and **4**, which use Br as the bridging halogen.

The most striking difference is that there is a twist between the two $Ag_{12}M$ structures (M = Pt or Pd) in **3** (dihedral angles $\theta = 9.02 - 11.85°$) and **4** ($\theta = 9.90 - 12.97°$), unlike in **1** and **2** (both $\theta = 0°$) (Fig. 2a–d and S9). Br$^-$ (1.95 Å)[39] has a larger ionic radius than Cl$^-$ (1.81 Å)[39], and the Ag−Br bond (2.619−2.659 Å for **3**) has a longer bond length than the Ag−Cl bond (2.444−2.532 Å for **1**) (Fig. S10). Therefore, if there is no twist in the two $Ag_{12}M$ structures (M = Pt or Pd) in **3** and **4**, the distance between the two $Ag_{12}M$ structures in those molecules should be longer than in **1** and **2** (Fig. S11). This would induce: (1) an increase in the distance between the shared Ag and the Ag bonded to it; and (2) a structural distortion of the individual $Ag_{12}M$ cores (Fig. S11), ultimately leading to the instability of the individual $Ag_{12}M$ structures (M = Pt or Pd). For **3** and **4**, it can be considered that the formation of such an unstable geometric structure is suppressed by twisting between the two $Ag_{12}M$ structures (M = Pt or Pd) (Fig. 2e and S12).

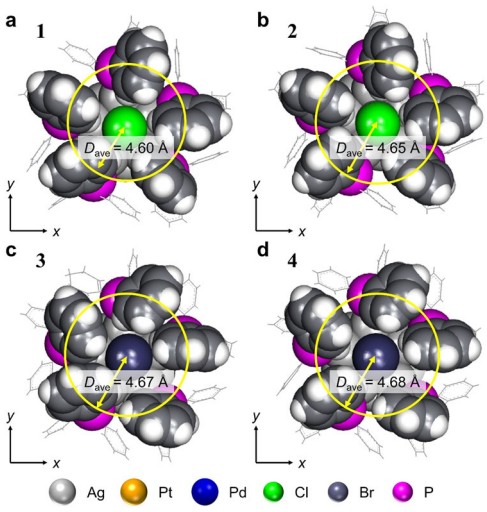

**Fig. 3 Structural analysis of the ligand positions. a–d** View from the long-axis direction for **1**, **2**, **3** and **4**, respectively, showing the average distance between the central long axis and the positions of the P atoms (Dave) (grey = Ag; orange = Pt; blue = Pd; green = Cl; dark grey = Br; magenta = P). The geometric structures of **1** and **2** are reproduced from ref. [28,31], respectively.

Regarding such superatomic molecules using Br as the bridging halogen, a similar twist between two icosahedral metal cores was not observed in $[Au_{23}Pd_2(PPh_3)_{10}Br_7]^0$ (**5**)[40] using Au as the base element, as reported by Zhu and colleagues (Fig. S13a). The Au—Br bond (2.569–2.583 Å for **5**) is shorter than the Ag—Br bond (2.619–2.659 Å for **3**) (Fig. S13b). Therefore, there may be no need for a twist between the two $Au_{12}Pd$ structures in **5** to preserve the individual icosahedral structures (Fig. S13c). Namely, in $[Au_{23}Pd_2(PPh_3)_{10}X_7]^0$ (X = halogen), the distance between the two $Au_{12}Pd$ structures is estimated to be relatively moderate when Br is used as a bridging halogen. Indeed, to the best of our knowledge, there have been no reports of the isolation of $[Au_{23}M_2(PPh_3)_{10}Cl_7]^0$ (M = Pt or Pd) using Cl, which has a smaller ionic radius to Br, as the bridging halogen. It is assumed that $[Au_{23}M_2(PPh_3)_{10}Cl_7]^0$ (M = Pt or Pd) is difficult to isolate because the distance between the two $Au_{12}M$ structures is too small, and it is excessively structurally stressful for the individual $Au_{12}M$ structures.

A second notable difference is that the $PPh_3$ structure is located slightly further from the long axis of the superatomic molecule in **3** and **4** compared with in **1** and **2** (Fig. 3). Because Br has a larger ionic radius than Cl, there might be a slight steric hindrance between the terminal Br and $PPh_3$ in **3** and **4**. This seems to produce less variation in the spread angle of the long axis and $PPh_3$ of the superatomic molecule (Fig. S14a), and the length of the Ag—P (P = phosphorus) bond (Fig. S14b) in **3** and **4** compared with in **1** and **2**. Note that **3** has more variation in Ag—Ag bond length than **1** (Fig. S15).

Therefore, the type of bridging halogen induces slight differences in the geometric structures of the obtained superatomic molecules. However, in all of **1**–**4**, $Ag_{12}M$ structures (M = Pt or Pd) is bridged by five halogens, which is common to all of the geometric structures in **1**–**4**.

Meanwhile, in previous work on the connection of $Au_7Ag_6$, Teo et al. successfully synthesised $[Au_{13}Ag_{12}(P(p\text{-}Tol)_3)_{10}Cl_7](SbF_6)_2$ (P(p-Tol)₃ = tri(p-tolyl)phosphine; $SbF_6^-$ = hexafluoroantimonate; **6**)[41] bridged by five Cl atoms, and $[Au_{13}Ag_{12}(PPh_3)_{10}Cl_8](SbF_6)$ (**7**)[42] bridged by six Cl atoms. In addition, when Br was used as the bridging halogen, they successfully synthesised $[Au_{13}Ag_{12}(PPh_3)_{10}Br_8](SbF_6)$ (**8**)[43], $[Au_{12}Ag_{13}(P(p\text{-}Tol)_3)_{10}Br_8]$ (PF₆) ($PF_6^-$ = hexafluorophosphonate; **9**)[44], and $[Au_{13}Ag_{12}(PPh_3)_{10}Br_8]Br$ (**10**)[45] bridged by six Br atoms, and even

$[Au_{13}Ag_{12}(PMePh_2)_{10}Br_9]^0$ (PMePh₂ = methyldiphenylphosphine; **11**)[46] bridged by seven Br atoms. Although **7**–**11** are connected by a different number of bridging halogens from **1**–**4** and **6** (Fig. S16 and Table 1), the total number of valence electrons is estimated to be 16 in all cases[16]. Therefore, **7**–**11** are also considered to be a di-superatomic molecule with two $Au_7Ag_6$ or $Au_6Ag_7$ structures connected by vertex sharing. However, in the present study, the formation of superatomic molecules bridging two $Ag_{12}M$ structures (M = Pt or Pd) with six or seven halogens (X = Cl or Br), such as $[Ag_{23}M_2(PPh_3)_{10}X_8]^-$ (the total number of valence electrons = 16) or $[Ag_{23}M_2(PPh_3)_{10}X_9]^{2-}$ (the total number of valence electrons = 16), was not observed. These anions would be readily oxidised under atmospheric conditions[47,48], leading to a change in the total number of valence electrons of $[Ag_{23}M_2(PPh_3)_{10}X_8]^0$ and $[Ag_{23}M_2(PPh_3)_{10}X_9]^0$ from 16[16] to 15 or 14, respectively. In these cases, the individual $Ag_{12}M$ structures do not necessarily have closed-shell electronic structures. This explains why $[Ag_{23}M_2(PPh_3)_{10}X_8]^-$ and $[Ag_{23}M_2(PPh_3)_{10}X_9]^{2-}$ were not produced in our study. Similarly, Teo et al. only reported the formation of $[Au_{11}Ag_{12}Pt_2(PPh_3)_{10}Cl_7]^0$ (**12**) bridged by five Cl atoms for a superatomic molecule with Pt at the centre of the metal core[49]. Kappen et al. also only reported $[Au_{10}Ag_{13}Pt_2(PPh_3)_{10}Cl_7]^0$ (**13**)[50] bridged by five Cl atoms for superatomic molecules containing Pt at the centre of the metal core. It is assumed that $[Au_{11}Ag_{12}Pt_2(PPh_3)_{10}Cl_7]^-$ and $[Au_{10}Ag_{13}Pt_2(PPh_3)_{10}Cl_6]^-$ could not be isolated in their study for the same reason.

**Electronic structure.** Figure 4a–d shows the optical absorption spectra of dichloromethane solutions of **1**–**4**, respectively. The optical absorption spectra are generally similar in shape, but the peak structure shifts to a longer wavelength when the central atom is changed from Pt to Pd.

Both **1** and **2** belong to the $D_{5h}$ point group[28,31,32]. Based on the calculated electronic structures of $[Ag_{23}Pt_2(PPh_3)_{10}Cl_7]^0$ (**1′**) and $[Ag_{23}Pd_2(PPh_3)_{10}Cl_7]^0$ (**2′**), the peak of the first absorption band on the longer wavelength side is attributed to an allowed transition between the orbitals originated from the core ($a_2'' \rightarrow a_1'$) (Fig. 5)[28,31,32]. The second peak that appears on the shorter wavelength side in the absorption spectrum is attributed to a charge transfer transition from the $a_2''$ orbital originating from the core to the orbital with charge distribution around $PPh_3$ (Table S2). With regard to the change in peak position due to the difference in the central atom, our previous studies have shown that changing the central atom from Pt to Pd causes a red shift in the peak structure due to a decrease in the energy of the orbitals near the lowest unoccupied molecular orbital (LUMO)[28].

We also performed DFT calculations for **3** and **4** in the present study. The geometric structures ($[Ag_{23}Pt_2(PPh_3)_{10}Br_7]^0$ (**3′**; Fig. S17) and $[Ag_{23}Pd_2(PPh_3)_{10}Br_7]^0$ (**4′**; Fig. S18)), and the electronic structures (Fig. S19) calculated using the PBE functional both reproduced the experimental results well. Both **3′** and **4′** belong to the $D_5$ point group. Based on the calculated electronic structures of **3′** and **4′**, the peak of the first absorption band on the longer wavelength side is attributed to an allowed transition between orbitals originating from the core ($a_2 \rightarrow a_1$) (Fig. 5). The second peak that appears on the shorter wavelength side is attributed to a charge transfer transition from the $a_2$ orbital originating from the core to the orbital with charge distribution around $PPh_3$ (Table S2). There was no significant difference in the energy of the highest occupied molecular orbital (HOMO) between **3′** and **4′**, and the HOMOs were similar in energy compared with those of **1′** and **2′**. However, the orbital ($a_1$) energy on the LUMO side was much lower in **4′** than in **3′** (Figs. 4e, f, 5c, d). The red shift in the peak

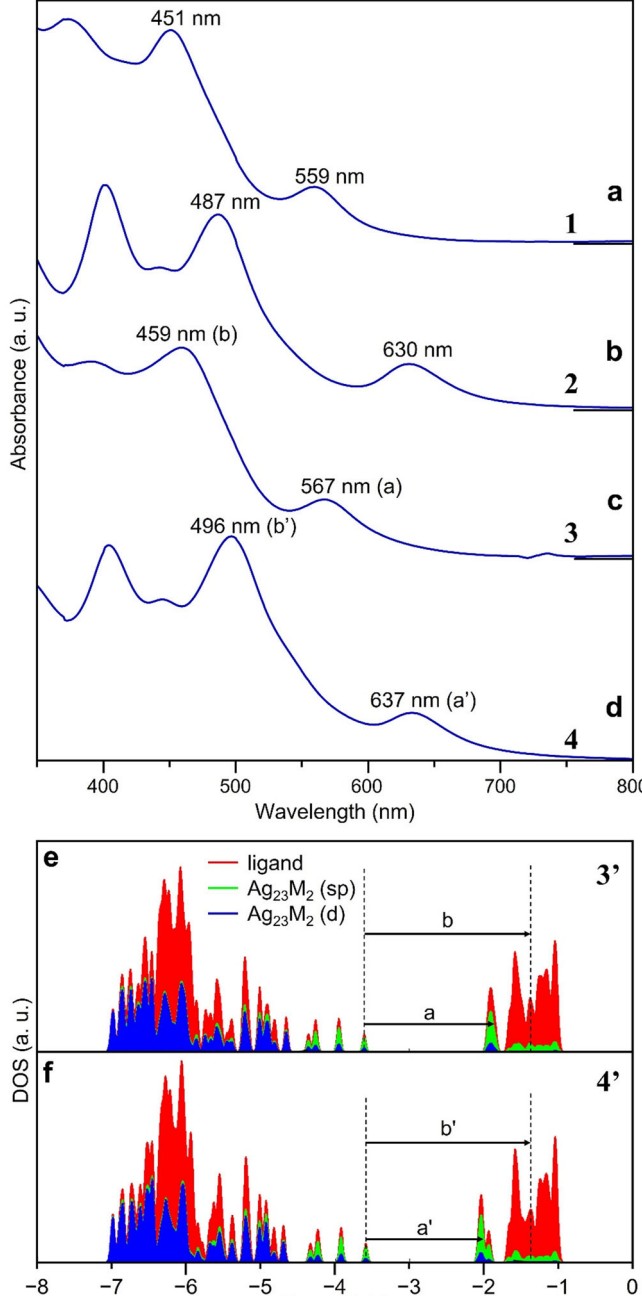

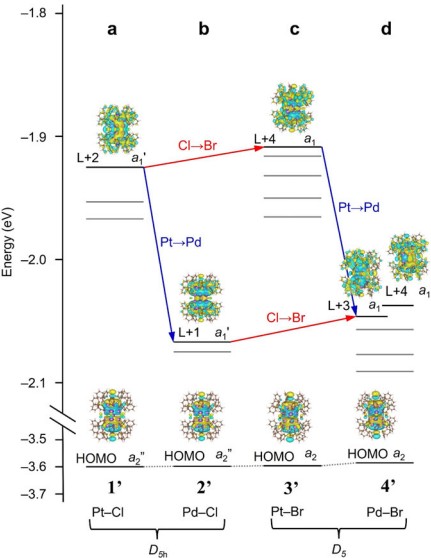

**Fig. 5 Orbital energies and Kohn–Sham orbital diagram related to the first peak in the optical absorption spectrum. a**, **b**, **c**, and **d** are Kohn–Sham orbital diagram of **1′**, **2′**, **3′**, and **4′**, respectively. The transition dipole moment from HOMO to LUMO becomes zero. This is the reason why the HOMO−LUMO transition is forbidden for **1**−**4**.

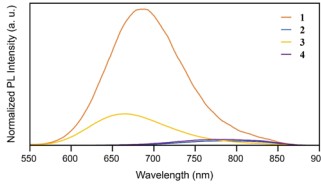

**Fig. 6 PL spectra obtained for the toluene solution of 1−4 at 25 °C.** The toluene solutions of **1**−**4** were excited by the light of 451, 487, 459 and 496 nm, respectively (orange = **1**, dark blue = **2**, yellow = **3**, purple = **4**). In this figure, the vertical axis is normalised to eliminate the effect of the difference of the concentration of the **1**−**4** on the PL intensity.

**Fig. 4 Optical absorption spectra and analyses. a**–**d** Optical absorption spectra of **1**, **2**, **3** and **4**, respectively. **e**, **f** Density of states of **3′** and **4′**, respectively. In **e** and **f**, a, b, a′ and b′ correspond to the peaks labelled as such in **c** and **d** (red = ligand; green = Ag₂₃M₂ (sp); blue = Ag₂₃M₂ (d)).

structure caused by the change of the central atom from Pt to Pd was found to be due to these factors, similar to the case when Cl is used as the bridging halogen. We have also investigated the possibility that the red shift in the peak structure is caused by the twist due to the use of Br as a bridging halogen. Specifically, we have calculated the optical absorption spectra also for $[Ag_{23}M_2(PPh_3)_{10}Br_7]^0$ (M = Pt or Pd) without distortion (Figs. S20, S21). The results demonstrated that the optical absorption spectrum changes only a little depending on the twist, supporting the above interpretation that the main reason for the red shift in the peak structure is the change of the central atom from Pt to Pd.

For **1**−**4**, it is difficult to estimate the HOMO−LUMO gap of each superatomic molecule from its optical absorption spectrum because the HOMO−LUMO transition is forbidden (Fig. 5). Therefore, we estimated the HOMO−LUMO gap of each di-superatomic molecule based on the **1′**−**4′** electronic structure obtained by DFT calculations. As a result, **1′**−**4′** were estimated to have HOMO−LUMO gaps of 1.66, 1.55, 1.66 and 1.52 eV, respectively (Table S3). These results indicate that the change of the central atom from Pt to Pd also induces a decrease in the HOMO−LUMO gap. Although we also attempted to conduct the electrochemical experiment[11] to experimentally determine the HOMO−LUMO gap, unfortunately, we could not obtain a reliable voltammogram due to the lack of the required amount of the obtained crystals.

Regarding the electronic structure, we also measured PL spectra of **1**−**4** (Fig. 6). The results demonstrated that (1) **1**−**4** exhibit PL in the visible-to-NIR region and (2) PL peak positions of **2** and **4** are red-shifted compared to those of **1** and **3**. This trend is well consistent with that of optical absorption, implying that HOMO and LUMO regions (Fig. 5) are related to the PL of **1**−**4**.

**Stability**. We investigated the stabilities of **1**−**4** with regard to degradation in toluene or dichloromethane solution by optical

absorption spectroscopy. Ag NCs generally have low photostability[31]. Furthermore, in the present study, we also dealt with less stable and less easily formed superatomic molecules to clarify the necessary conditions for the formation of a superatomic molecule composed of $Ag_{13-x}M_x$. Therefore, the solutions were kept in the dark during the stability measurements.

None of the superatomic molecules was very stable in the dichloromethane solution, and the shapes of their spectra changed dramatically over time (Fig. S22). Figure 7a–d shows the time-dependent changes of the optical absorption spectra of the toluene solutions of 1−4, respectively. As shown in Fig. 7a, 1 was quite stable in the toluene solution for three days. On the other hand, the shapes of the spectra of 2−4 changed gradually over time (Fig. 7e). We found that 2 and 4 were particularly unstable. Figure 7 demonstrates that the stability of 1−4 decreases in the order 1 > 3 > 2 > 4. In the present study, we also attempted synthesis using only $AgNO_3$ as the metal salt. The results demonstrated that somethings were synthesised just after adding $NaBH_4$ into the solution even when the precursor salt of heteroatoms ($H_2PtCl_6$, $Pd(PPh_3)Cl_2$, $PtBr_2$ or $PdBr_2$) was not included in the solution: the solution colour became yellow just after adding $NaBH_4$ into solution. However, the solution soon became colourless and the black precipitate was obtained. According to these results, it can be considered that the stability of $[Ag_{25}(PPh_3)_{10}X_7]^{2+}$ (X = Cl or Br) is quite low even if those clusters could be formed in solution. Similar results were reported by ref. [31]. Taking into account all the results mentioned above, the superatomic molecules described by $[Ag_{23}M_2(PPh_3)_{10}X_7]^z$ (M = Ag, Pd, or Pt; X = Cl or Br; z = 2+ or 0) are interpreted to decrease in stability in the order $[Ag_{23}Pt_2(PPh_3)_{10}Cl_7]^0$ (1) > $[Ag_{23}Pt_2(PPh_3)_{10}Br_7]^0$ (3) > $[Ag_{23}Pd_2(PPh_3)_{10}Cl_7]^0$ (2) > $[Ag_{23}Pd_2(PPh_3)_{10}Br_7]^0$ (4) (which are experimentally synthesisable) > $[Ag_{25}(PPh_3)_{10}X_7]^0$ (X = Cl or Br; which are not so stable in solution).

**Key factors for formation and isolation**. The substitution of the central atom of each icosahedral core by Pt or Pd is very effective for forming a superatomic molecule consisting of two $Ag_{13-x}M_x$ structures (M = Ag or other metal) connected by vertex sharing. Based on the DFT calculations by Baraiya et al., the Pt or Pd substitution of the central atom of $Ag_{13}$ leads to an increase in the average binding energy in the NCs[37]. The fact that it was possible to generate 1−4, whereas $[Ag_{25}(PPh_3)_{10}X_7]^{2+}$ (X = Cl or Br) was difficult to isolate, seems to be largely related to the individual icosahedral cores of 1−4 being stronger than those of $[Ag_{25}(PPh_3)_{10}X_7]^{2+}$ (X = Cl or Br) owing to the increase in the average binding energy. The fact that $[Ag_{23}Pt_2(PPh_3)_{10}X_7]^0$ is more stable than $[Ag_{23}Pd_2(PPh_3)_{10}X_7]^0$ (X = Cl or Br) can also be explained by the difference in average binding energy.

Regarding these heteroatomic substitutions of the central atom, Kang et al. pointed out that (1) they also affect the charge state of Ag in the L4 and L6 layers in Fig. S23a; and (2) without central atom substitution, $[Ag_{25}(PPh_3)_{10}Cl_7]^{2+}$ would not form stably owing to high charge repulsion between the L4 and L6 layers[40]. We therefore estimated the natural charges[51,52] of Ag in the L4 and L6 layers for 1′−4′, and $[Ag_{25}(PPh_3)_{10}Cl_7]^{2+}$ (5′; Table 1) and $[Ag_{25}(PPh_3)_{10}Br_7]^{2+}$ (6′). The results showed no strong correlation between the charge repulsion in the L4−L6 layer and the stability of 1−4 and $[Ag_{25}(PPh_3)_{10}X_7]^{2+}$ (X = Cl or Br): the magnitude of charge repulsion was estimated to be in the order 1′ > 5′ > 2′ = 3′ > 4′ = 6′ (Fig. S23b) and this order does not consistent with that of the stability (1 > 3 > 2 > 4). These results suggest that, although central atom substitution certainly affects the charge state of Ag in the L4 and L6 layers, such charge repulsion is not the main reason of the fact that $[Ag_{25}(PPh_3)_{10}X_7]^{2+}$ (X = Cl or Br) is difficult to isolate.

The present study also revealed that a superatomic molecule consisting of two $Ag_{13-x}M_x$ structures can be formed even when Br is used as the bridging halogen instead of Cl. As mentioned above, in 3 and 4, the twist between the two $Ag_{12}M$ structures (Fig. 2c, d) prevents each $Ag_{12}M$ structure from becoming unstable. Based on the results obtained in the present study, the type of bridging halogen appears to have little effect on whether superatomic molecules can be formed or not, as long as the bridging halogen is large enough to maintain a moderate distance between the two $Ag_{12}M$ structures.

It should be noted that the type of bridging halogen has a slight effect on the binding energy of the Ag−X bond. That 1 is slightly more stable than 3, and 2 is slightly more stable than 4 may be related to the Ag−Cl bond (314 kJ $mol^{-1}$) being stronger than the

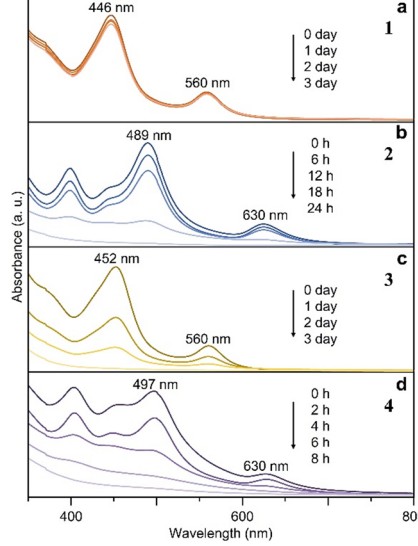

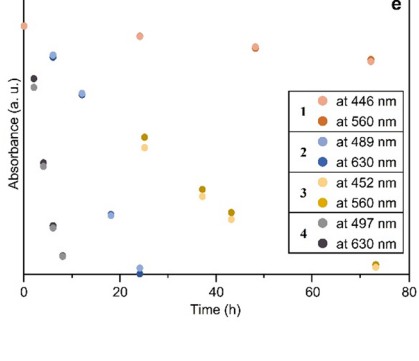

**Fig. 7 Time dependence of the optical absorption spectra. a–d** are time-dependent optical absorption spectra of **1**, **2**, **3** and **4**, respectively. **e** Time dependence of plots of absorbance of the first and second peaks in the optical absorption spectra of **1** (446 and 560 nm), **2** (489 and 630 nm), **3** (452 and 560 nm) and **4** (497 and 630 nm) (orange = **1**, dark blue = **2**, yellow = **3**, purple = **4**). In these spectra, the peak positions are little shifted compared to those in Fig. 4, probably due to the difference in solvent (dichloromethane for Fig. 4 vs. toluene for Fig. 7).

Ag−Br bond (293 kJ mol$^{-1}$)[53]. In addition, the type of bridging halogen also has a slight effect on the variation in Ag−Ag bond length within each Ag$_{12}$M molecules (Fig. S15). These results suggest that the type of bridging halogen affects the stability depending on the binding energy of the Ag−halogen bond and the variation in Ag−Ag bond length within each Ag$_{12}$M molecules.

In [Ag$_{23}$M$_2$(PPh$_3$)$_{10}$X$_7$]$^z$, in addition to the bridging sites, the halogen is also coordinated at both ends of the long axis of the superatomic molecule. Therefore, the type of halogen also affects the length of the Ag−P bond (Fig. 3), and consequently, for example, some of the Ag−P bonds are longer in 1 (Fig. S14b). This means that some Ag−P bonds are more easily dissociated in [Ag$_{23}$Pt$_2$(PPh$_3$)$_{10}$Cl$_7$]$^0$. However, as mentioned above, 1 is the most stable against degradation among 1−4. These results indicate that the slight difference in Ag−P bond length caused by the difference in the halogen species at both ends in the superatomic molecule does not determine the stability order of [Ag$_{23}$M$_2$(PPh$_3$)$_{10}$X$_7$]$^z$, although the detachment of PPh$_3$ seems to be also included in the degradation of the superatomic molecules, as shown in Fig. 7 (Fig. S24).

Finally, in the present study, we were only able to confirm the formation of superatomic molecules with five bridging halogens. This is considered to be largely because when the number of bridging halogens ($y − 2$) is higher than five in [Ag$_{25−x}$M$_x$(PR$_3$)$_{10}$X$_y$]$^z$ (M = Pt or Pd; X = Cl or Br; $y$ = number of X), the total number of valence electrons is 16 only if the molecule is an anion. Anions are generally not highly resistant to oxidation in air[47,49]. These results suggest that isolatable [Ag$_{25−x}$M$_x$(PR$_3$)$_{10}$X$_y$]$^z$ must have a substitutional heteroatom species and a certain number of bridging halogens such that the total number of valence electrons is 16 in the cationic or neutral state (Table 1).

The factors discussed above suggest that the following three conditions are required to stabilise superatomic molecules consisting of two Ag$_{13−x}$M$_x$ structures (M = Ag or other metal) connected by vertex sharing ([Ag$_{25−x}$M$_x$(PR$_3$)$_{10}$X$_y$]$^z$): (1) a halogen of sufficient size to maintain a moderate distance between the two Ag$_{13−x}$M$_x$ structures is used as the bridging halogen (Fig. 8a); (2) an icosahedral core, which is stronger than Ag$_{13}$, is formed by heteroatom substitution (Fig. 8b); and (3) the combination of the substituted heteroatoms and the number of bridging halogens is such that the total number of valence electrons is 16 when the molecule is cationic or neutral (Fig. 8c). For (1), halogens with ionic radii equal or larger than that of Cl fall into this category, and for (2), the central atom substitution with Pt or Pd satisfies this condition. Based on the reports by refs. 41−46, condition (2) is also satisfied by multiple atom

substitution with Au[37]. For (3), the number of bridging halogens ($y − 2$) is limited to 5 when Pt or Pd is the heteroatom, but when Au is the heteroatom[41−46], the number of bridging halogens can be in a range of 5 to 7 (Fig. S16). As long as these three essential conditions are met at the same time, it is possible to stabilise and thereby isolate a superatomic molecule with two Ag$_{13−x}$M$_x$ structures connected by vertex sharing. To increase the stability of the resulting superatomic molecule, it is preferable to use Cl as the bridging halogen, and to combine multiple heteroatomic substitutions to stabilise the metal core. Therefore, it is assumed that 12 and 13 are even more stable than 1. Moreover, it is expected that it will be possible to isolate [Ag$_{23}$PtPd (PPh$_3$)$_{10}$X$_7$]$^0$, [Ag$_{23}$Ni$_2$(PPh$_3$)$_{10}$X$_7$]$^0$, [Ag$_{23}$PtNi(PPh$_3$)$_{10}$X$_7$]$^0$, and [Ag$_{23}$PdNi(PPh$_3$)$_{10}$X$_7$]$^0$ (X = Cl or Br) superatomic molecules in the future[20,37,54]. According to our experiment on the stability (Fig. S24), the addition of an excess PPh$_3$ to the solution seems to help isolate new superatomic molecules. So far, we have only discussed cases in which PR$_3$ and X are used as ligands. However, if recently reported multidentate ligands[29,30] are used as bridging ligands, it may be possible to create even more types of superatomic molecules, such as those consisting of two Ag$_{13}$ structures connected by vertex sharing. The knowledge obtained in this study is expected to be also useful to stabilise and thereby isolate the longer superatomic molecules composed of three or four superatoms.

Although the present study is concerned with superatomic molecules composed of Ag$_{13−x}$M$_x$ (M = Ag or other metal), the above-mentioned conditions (1−3) also seem to be requirements for stabilising and thereby isolating superatomic molecules consisting of two Au$_{13−x}$M$_x$ structures (M = Ag or other metal) connected by vertex sharing. The point of difference from the case of Ag$_{13−x}$M$_x$ is the threshold in (1). In the case of Au$_{13−x}$M$_x$, the halogens with ionic radii equal or larger than that of Br are assumed to fall under condition 1[40]. Because Au differs from Ag in its formation of strong bonds with thiolate (SR) and selenolate (SeR)[55−57], for superatomic molecules composed of Au$_{13−x}$M$_x$, even more stable superatomic molecules can be obtained if SR and SeR are used as bridging ligands[58]. In fact, it has been reported that [Au$_{25}$(PPh$_3$)$_{10}$(SR)$_5$Cl$_2$]$^{2+}$ (SR = alkanethiolate[21] or PET[59]) and [Au$_{25}$(PPh$_3$)$_{10}$(SePh)$_5$Br$_2$]$^{+/2+}$ (SePh = phenylselenolate)[60] connected with Au$_{13}$ without heteroatom substitution can also be isolated when SR or SeR is used as the bridging ligand. We have also successfully isolated [Au$_{24}$Pd(PPh$_3$)$_{10}$(PET)$_5$Cl$_2$]$^+$ in which Pd substitution occurred only in one icosahedral core[61]. It is expected that such superatomic molecules connected with Au$_{13−x}$M$_x$ by SR or SeR, [Au$_{24}$Pt(PR$_3$)$_{10}$(SR)$_5$]$^+$ and [Au$_{23}$PtPd(PR$_3$)$_{10}$(SR)$_5$]$^0$ (PR$_3$ = PPh$_3$, P($p$-Tol)$_3$[41,43,45] or PMePh$_2$[46]), will be isolated in the future.

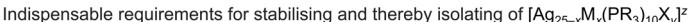

Indispensable requirements for stabilising and thereby isolating of [Ag$_{25−x}$M$_x$(PR$_3$)$_{10}$X$_y$]$^z$

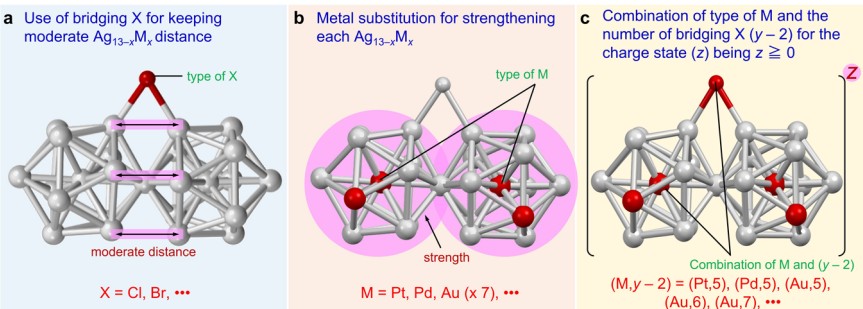

**Fig. 8 Indispensable requirements for stabilising and thereby isolating [Ag$_{25−x}$M$_x$(PR$_3$)$_{10}$X$_y$]$^z$. a** Use of a bridging X with a relevant ion radius (grey = Ag or M; red = bridging X). **b** Metal substitution for strengthening each Ag$_{13−x}$M$_x$ (grey = Ag; red = M). **c** Combination of types of M and the number of X ($y − 2$) for a charge state ($z$) with high resistance to oxidation (grey = Ag; red = M or bridging X).

## Methods

### Synthesis

$[Ag_{23}Pt_2(PPh_3)_{10}Br_7]^0$ **(3)**. All syntheses were performed at 25 °C. First, 30 mg (0.18 mmol) of $AgNO_3$ and 5.1 mg (0.05 mmol) of NaBr were dissolved in 5 mL of methanol, and then 5 mL of methanol containing 2.1 mg (0.006 mmol) of $PtBr_2$ was added to the solution. The mixed solution was stirred for 15 min, and then 30 mL of methanol containing 262 mg (1 mmol) of $PPh_3$, which was sonicated to disperse in methanol, was added. After stirring for 15 min, 1 mL of a methanol solution containing 20 mg (0.529 mmol) of $NaBH_4$ was rapidly added to the solution and the resulting solution was stirred for another 24 h. All experiments up to this point were performed in the dark. The solvent was then removed from the solution by rotary evaporation. Then, toluene was added to extract the product, and then water was added to the solution. After centrifugation, the toluene layer was separated to eliminate the excess $NaBH_4$, and the solvent of the solution was evaporated using an evaporator to obtain the desired NC (**3**) (Fig. S25) (See S1.1 for chemicals). The chemical composition was confirmed by ESI-MS (Fig. S2), XPS (Fig. S3) and SC-XRD (See S1.2 for crystallographic method).

$[Ag_{23}Pd_2(PPh_3)_{10}Br_7]^0$ **(4)**. First, 30 mg (0.18 mmol) of $AgNO_3$ was dissolved in 5 mL of methanol, and then 5 mL of methanol containing 1.6 mg (0.006 mmol) of $PdBr_2$ was added to the solution. After 15 min of stirring, 30 mL of methanol containing 262 mg (1 mmol) of $PPh_3$, which was sonicated to disperse in methanol, was added to the solution. After stirring for 15 min, 1 mL of a methanol solution containing 20 mg (0.529 mmol) of $NaBH_4$ was rapidly added to the solution and the resulting solution was stirred for another 24 h. All experiments up to this point were performed at 0 °C in the dark. Note that, unlike in the synthesis of **3**, it was not necessary to increase the quantity of Br ions in the solution by adding TOABr during the synthesis of **4**. The solvent was then evaporated from the mixed solution using a rotary evaporator. Then, toluene was added to extract the product, and then water was added to the solution. After centrifugation, the toluene layer was separated to eliminate the excess $NaBH_4$, and the solvent of the solution was evaporated using an evaporator to obtain the desired NC (**4**) (Fig. S26) (See S1.1 for chemicals). SC-XRD (See S1.2 for crystallographic method) was used for confirming the geometry and composition of **4** except Pd atoms, which were confirmed by XPS (Fig. S4) and ICP-MS. ESI-MS of **4** was not succeeded owing to the instability of **4**.

**Crystallisation**. Compounds **3** and **4** were crystallised using the liquid−liquid diffusion method. **3** or **4** was first dissolved in ethanol and the solution was placed in a crystallisation vial. Six equivalent amount of hexane was then gently placed on the ethanol solution of **3** or **4**. The crystallisation vial was covered with a lid and the vial was allowed to stand at 25 °C. Orange needle-like crystals were obtained after a few days.

**Characterisation**. ESI-MS was performed with an ESI-Qq-TOF-MS compact (Bruker, MA, USA). In the experiment, first, multiple crystals of **3** were dissolved in toluene with $PPh_3$ (1 mM), which suppresses the detachments of $PPh_3$ from the superatomic molecules in the solution (Fig. S24). Then, methanol was added to this solution (toluene:methanol = 3 : 1 (v/v)). Finally, 5 mM caesium carbonate ($Cs_2CO_3$) methanol solution was added to the solution. The obtained solution was electrosprayed at a flow rate of 200 μL/min.

The SCs were immersed in cryoprotectant Parabar 10312 (Hampton Research, California, USA) and mounted on a MicroLoops E Inclined Assortment™ (MiTeGen, New York, USA). The SC-XRD data sets were collected in a Bruker D8 QUEST, using monochromated MoKα radiation ($\lambda = 0.71073$ Å). Bruker Apex 3[62] suite was used for solving preliminary structures by following the sequential steps: indexing, data integration, reduction, absorption correction (multi-scan), space group determination and structure solution (with the intrinsic-phasing method). Final refinement was performed by *SHELXL*-2018/3[63] using the *Olex*2 platform[64] (Tables S1, S2).

The optical absorption spectra of the dichloromethane solutions of **3** and **4** were obtained at 25 °C using a V-630 spectrometer (JASCO, Tokyo, Japan). Multiple crystals were dissolved in dichloromethane for the measurement.

PL spectra of the toluene solution of **1**−**4** were measured using an FP-6300 spectrofluorometer (JASCO, Tokyo, Japan) at 25 °C. PL intensity ($F_{nor.}(\lambda)$) was normalised using the following equation to eliminate the effect of the difference in the concentration of **1**−**4** on the PL intensity.

$$F_{nor.}(\lambda) = F(\lambda_{em})/[1-10^{-A(\lambda ex)}]$$

Where $\lambda_{em}$, $\lambda_{ex}$, $A$ and $F$ represent the emission wavelength, excitation wavelength, absorbance and PL intensity, respectively.

XPS spectra were collected using a JPS-9010MC electron spectrometer (JEOL, Tokyo, Japan) at a base pressure of $\sim 2 \times 10^{-8}$ Torr. X-rays from the Mg-Kα line (1253.6 eV) were used for excitation. An indium plate was used as a substrate. The spectra were calibrated with the peak energies of In 3d$_{3/2}$ (451.2 eV)[65].

**Stability experiments**. To investigate the stability of **1**−**4** with regard to decomposition in solution, solutions of **1**, **2**, **3** or **4** were prepared and measured in the following three different ways.

1. Dichloromethane solutions of each sample were placed in the glass cell of a spectrophotometer at 25 °C. The optical absorption spectrum of each solution was regularly measured for 1 h (Fig. S22).
2. Toluene solutions of each sample were left in a test tube with a lid at 30 °C. The optical absorption spectra were measured regularly for 3 days of the solutions of **1** and **3**, 1 day of the solution of **2**, and 8 h of the solution of **4**. (Fig. 7).
3. Toluene solutions with $PPh_3$ (95 mM) of each sample were placed in the glass cell of a spectrophotometer at 25 °C. The optical absorption spectrum of each solution was regularly measured for 1 week (Fig. S24).

**DFT calculations**. We performed DFT calculations on **3′**, **4′** and **6′** using the structures of the experimentally synthesised **3** and **4**; Pd was replaced with Ag for the calculation of **6′**. All DFT calculations were performed with TURBOMOLE[66] under the resolution of identity approximation with the PBE[67] functional using the def-SV(P) basis sets[68] along with the relativistic effective core potentials for Pd, Ag, and Pt[69]. Optimised structures with different Pd positions were obtained at the same level of theory. The electronic absorption spectra were simulated in the framework of time-dependent DFT[70–73], in which the line spectra were convoluted by a Lorentz function with a width of 10 nm. PBE was used as a function to calculate the absorption spectrum. Optimised structures with different Pd positions (**3′**) and absorption spectra (**3′**, **4′** and **6′**) were also calculated using CAM-B3LYP as a functional (Figs. S27, S28), which produced similar overall results to those obtained using PBE as a functional.

## Data availability

The X-ray crystallographic coordinates for structures reported in the present study have been deposited at the Cambridge Crystallographic Data Centre (CCDC), under deposition numbers 2195306−2195307. These data can be obtained free of charge from The Cambridge Crystallographic Data Centre via www.ccdc.cam.ac.uk/data_request/cif. cif and cif check of **3** and **4** are provided in supplementary Data 1–4. The atomic coordinates of the DFT-optimised structures of **1′-4′**, **6′** have been provided in Supplementary Data 5–9, respectively. All other data are available from the corresponding authors on reasonable request.

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

## Acknowledgements

We thank Mai Ishimi, Aoi Akiyama and Miyu Sera (Tokyo University of Science) for their support on the experiments. This work was supported by the Japan Society for the Promotion of Science (JSPS) KAKENHI (grant numbers 20H02698 and 20H02552), Scientific Research on Innovative Areas "Innovations for Light-Energy Conversion" (grant numbers 18H05178 and 20H05115), Scientific Research on Innovative Areas "Hydrogenomics" (grant number 21H00027) and Scientific Research on Innovative Areas "Aquatic Functional Materials" (grant numbers and 22H04562). Funding provided by the Yazaki Memorial Foundation for Science and Technology, the Ogasawara Foundation for the Promotion of Science and Engineering, and TEPCO Memorial Foundation, the Japan Science Society, the Takahashi Industrial and Economic Research Foundation and the Kubota Corporation is also gratefully acknowledged.

## Author contributions

Y.N. and S.H. designed the experiments and conducted the measurements along with S.M., A.I., T. Kosaka, Y.N. and T. Kawawaki. T.I. and T.T. performed the DFT calculations. Y.N. and S.H. wrote the paper. All authors have approved the final version of the manuscript.

## Competing interests

The authors declare no competing interests.
