## [Peer Review File · Communications Chemistry]

Reviewers' comments:

Reviewer #1 (Remarks to the Author):

This manuscript reports the synthesis and characterization of two new 16-electron superatomic molecules (compounds 3 and 4) that differ from the authors' previously reported compounds 1 and 2 only by the nature of the halogen ligand (Br instead of Cl). The structures of these four clusters are analyzed in detail and compared to those of a series of closely related 16-electron species from the literature. This work is completed by a DFT computational investigation. The structural differences and relative stabilities of these clusters are rationalized according to their intimate composition. The work is well done and the manuscript is well written, but in my opinion its interest is limited to the narrow field of the bis-icosahedral 16-electron nanoclusters of the type of those listed in Table 1, with little opportunities of generalization. This is why I do not recommend publication in *Comms. Chem.* of this manuscript, which is rather intended to be published in a more specialized journal.

Additional remarks:

- 1) The first paragraph of the introduction contains only commonplaces for chemist readers.
- 2) Page 2, line 38. At least the two following references on the concept of superatomic molecules deserve to be cited: *J. Chem. Phys.* 2013, 138, 141101 and *Dalton Trans.* 2015, 44, 6680–6695.
- 3) Page 5, lines 84-91. The fact that M = Pd or Pt prefers occupying the center of a MAg₁₂ or MAu₁₂ icosahedron in 8-electron superatoms is now well established and has been rationalized (*J. Phys. Chem. C* 2019, 123, 9516–9527 and *Nanoscale*, 2021, 14, 196-203).
- 4) Page 9, line 144. This section does not describe the electronic structure of the clusters, but simply identifies the optical transitions. No information on the orbitals housing the 16 superatomic electrons, for instance.
- 5) Page 9, lines 157-159. It is written that 3 and 4 belong to the C_{5h} group, but no mention is made on the fact that 3' and 4' belong to the D_{5h} group (Figure S14). This is confusing. In addition, a₁ and a₂ are not irreducible representations of D_{5h}.

Reviewer #2 (Remarks to the Author):

It is of great interest and importance to study Ag-based superatomic molecules of metal nanoclusters, as they provide clear guidance for the creation of superatomic molecules with various properties and functions. Basset et al. designed and synthesized a rod-like, charge-neutral, diplatinum-doped Ag nanocluster of [Pt₂Ag₂₃Cl₇(PPh₃)₁₀] by a controlled doping strategy. In this work, based on the synthesized ([Ag₂₃Pt₂(PPh₃)₁₀Br₇]₀ and [Ag₂₃Pd₂(PPh₃)₁₀Br₇]₀), the effects of central atom type and bridging halogen on the electronic structures of the resulting superatomic molecules are discussed. The effects of halogen atoms and bridging ligands on the formation of stable supramolecular molecules are also demonstrated. The manuscript is well written and it is suggested to be revised for publication. There are some suggestions.

1. Although the UV-vis spectra and XPS spectra of the [Ag₂₃Pt₂(PPh₃)₁₀Br₇]₀ and [Ag₂₃Pd₂(PPh₃)₁₀Br₇]₀ nanoclusters were performed, more characterizations such as ESI, TGA, etc. are desirable to demonstrate the purity of the nanoclusters for the experiments.
2. Whether the structure distortion caused by halogen atoms will affect the fluorescence intensity and emission location?
3. Please address or explain the Alert level A/B.
4. "For 1–4, it is difficult to estimate the HOMO–LUMO gap of each superatomic molecule from its optical absorption spectrum because the HOMO–LUMO transition is forbidden (Fig. S14)" (Line 155-156). Please give a more detailed explanation and add references.
5. Whether the HOMO-LUMO can be observed from the electrochemical spectra?

Reviewer #3 (Remarks to the Author):

Miyajima et al. reported the synthesis of two $\text{Ag}_{23}\text{Pt}_2(\text{PH}_3)_{10}\text{Br}_7/\text{Ag}_{23}\text{Pd}_2(\text{PH}_3)_{10}\text{Br}_7$ clusters, which own two icosahedral core. These two cores share a common metal atom and are bridged by halogen ligands. The authors then tried to clarify the effects of the central atom and the type of bridging halogen. Although this work was well organized, I cannot recommend the publication of this paper on communication chemistry because the existed limitations affects the rationality of this work. The reasons are given below.

1. This paper lacks the novelty to publish in a Nature Communication sister journal. As indicated in Table 1, several clusters of the similar structures have been synthesized before. The replacement of Br into Cl does not give us more understanding/guidance for future study.

2. The major discovery of this work indicates that halogen ligand with proper size is essential for the formation and isolation of a superatomic molecule consisting of two $\text{Ag}_{13-x}\text{M}_x$ structures ($\text{M}=\text{Ag}$ or other metal). Although not tried in this work, it cannot be totally excluded if thiolate ligand can achieve the same goal because Jin et al. ever prepared $[\text{Ag}_2\text{Au}_{23}(\text{PPh}_3)_{10}(\text{PET})_5\text{Cl}_2]^{2+}$ with the utilization of thiolate ligand. Thus, halogen ligand can be one of the practicable ways but not essential.

3. The authors state icosahedral core with heteroatom substitution is essential. However, to my understanding, the pure Ag cluster is still await to be prepared. I understand icosahedral core with heteroatom substitution can be more stable but this does not mean pure Ag cluster is impossible to be obtained. Because Bakr et al. ever obtained $[\text{Ag}_{25}(\text{SR})_{18}]^-$, which owns icosahedral core.

4. Line 56

A key flaw of this work. The author suggest the stability increases in the order $1>3>2>4$. However, it is hard to obtain this conclusion because the time dependence of the optical absorption spectra of 3 in Figure 5 shows the least variation to me. This may affect the discussions because substantial efforts were paid to rationalize the effects of different halogen ligands.

Other technique suggestions are given below.

1. Introduction

Several supporting figures (even in the first paragraph) were used to show the background of this work. To my understanding, this is not common. Instead, the authors can try to give citations in the introduction.

2. Table 1

1', 2', 3', and 4' can be removed because they indicate the same cluster with 1, 2, 3, and 4. p.w. can be used to indicate present work instead of '-'.

3. Line 95

The authors used the "ionic radius", but the cited reference actually means "atomic radius".

4. Line 202

How did the authors get the conclusion of "L4 and L6 layers do not own strong correlation". The corresponding Ag-Ag distance should be within 3.0 Å.

Reviewer: 1

Comment 1:

This manuscript reports the synthesis and characterization of two new 16-electron superatomic molecules (compounds 3 and 4) that differ from the authors' previously reported compounds 1 and 2 only by the nature of the halogen ligand (Br instead of Cl). The structures of these four clusters are analyzed in detail and compared to those of a series of closely related 16-electron species from the literature. This work is completed by a DFT computational investigation. The structural differences and relative stabilities of these clusters are rationalized according to their intimate composition. The work is well done and the manuscript is well written, but in my opinion its interest is limited to the narrow field of the bis-icosahedral 16-electron nanoclusters of the type of those listed in Table 1, with little opportunities of generalization. This is why I do not recommend publication in *Comms. Chem.* of this manuscript, which is rather intended to be published in a more specialized.

Reply 1:

We wish to thank the reviewer for providing valuable comments and suggestions. We have revised the manuscript in accordance with your comments. All the revised portions appear as red text in the revised manuscript.

To construct substances using superatoms and create new materials, **it is essential to gain a deeper understanding of the types of connected clusters that can be produced and the electronic structures that can be created.** However, in the previous studies, it has not been well understood about 1) which kinds of bi-icosahedral metal cluster can be produced and 2) what is the key point for connecting metal clusters. The synthesis of new bi-icosahedral metal clusters itself is not so great finding. However, we could reveal 1) and 2) by newly synthesizing $[\text{Ag}_{23}\text{Pt}_2(\text{PPh}_3)_{10}\text{Br}_7]^0$ and $[\text{Ag}_{23}\text{Pd}_2(\text{PPh}_3)_{10}\text{Br}_7]^0$ and comparing the stability of four kinds of clusters **1–4**. The obtained understanding well explains the reason why the reported bi-icosahedral metal clusters could be isolated, indicating that the key factor reported in this article is applicable not only to **1–4** but also to the other bi-icosahedral metal clusters. The elucidation of key factor leads to the prediction of the bi-icosahedral metal clusters which are expected to be isolated in the future, such as $[\text{Ag}_{23}\text{PtPd}(\text{PPh}_3)_{10}\text{X}_7]^0$, $[\text{Ag}_{23}\text{Ni}_2(\text{PPh}_3)_{10}\text{X}_7]^0$, $[\text{Ag}_{23}\text{PtNi}(\text{PPh}_3)_{10}\text{X}_7]^0$, and $[\text{Ag}_{23}\text{PdNi}(\text{PPh}_3)_{10}\text{X}_7]^0$ (X = Cl or Br) (line **256–258** in the revised manuscript). In this way, **the findings in this work are expected to provide clear design guidelines for the creation of novel materials with various properties and functions in the future. The obtained three information, 1)–3), should also be useful for the future work on the isolation of metal clusters composed of three or four superatoms.** Accordingly, we believe that the manuscript merits consideration as an Article in *Communications Chemistry*. Thus, please reconsider your conclusion. We have described the novelty and extendability of this work in the main manuscript (line **261–263** in the revised manuscript).

Comment 2:

The first paragraph of the introduction contains only commonplaces for chemist readers.

Reply 2:

We have removed the 1st paragraph of Introduction.

Comment 3:

Page 2, line 38. At least the two following references on the concept of superatomic molecules deserve to be cited: *J. Chem. Phys.* 2013, 138, 141101 and *Dalton Trans.* 2015, 44, 6680–6695.

Reply 3:

We have cited the suggested literatures (ref **18** and **19** in the revised manuscript).

Comment 4:

Page 5, lines 84-91. The fact that M = Pd or Pt prefers occupying the center of a MAg₁₂ or MAu₁₂ icosahedron in 8-electron superatoms is now well established and has been rationalized (*J. Phys. Chem. C* 2019, 123, 9516–9527 and *Nanoscale*, 2021, 14, 196-203).

Reply 4:

We have cited the suggested literatures (ref **34** and **35** in the revised manuscript).

Comment 5:

Page 9, line 144. This section does not describe the electronic structure of the clusters, but simply identifies the optical transitions. No information on the orbitals housing the 16 superatomic electrons, for instance.

Reply 5:

We have drawn the orbitals housing the 16 superatomic electrons in Figure S6 in the revised manuscript.

Fig. S6: Schematic of the orbital energies for Au_{13} superatom and Au_{25} superatomic molecule.

Comment 6:

Page 9, lines 157-159. It is written that 3 and 4 belong to the C_{5h} group, but no mention is made on the fact that 3' and 4' belong to the D_{5h} group (Figure S14). This is confusing. In addition, a1 and a2 are not irreducible representations of D_{5h} .

Reply 6:

We are sorry for this our careless mistake. We have redrawn the Figure S14 (Figure 5 in the revised manuscript).

Fig. 5: Orbital energies and Kohn-Sham orbital diagram related to the first peak in optical absorption spectrum. a 1'. b 2'. c 3'. d 4'. The overlap of the wave function between HOMO and LUMO is zero, therefore, the transition dipole moment from HOMO to LUMO becomes zero. This is the reason why the HOMO-LUMO transition is forbidden for 1-4.

Reviewer: 2

Comment 1:

It is of great interest and importance to study Ag-based superatomic molecules of metal nanoclusters, as they provide clear guidance for the creation of superatomic molecules with various properties and functions. Basset et al. designed and synthesized a rod-like, charge-neutral, diplatinum-doped Ag nanocluster of $[\text{Pt}_2\text{Ag}_{23}\text{Cl}_7(\text{PPh}_3)_{10}]^0$ by a controlled doping strategy. In this work, based on the synthesized $[\text{Ag}_{23}\text{Pt}_2(\text{PPh}_3)_{10}\text{Br}_7]^0$ and $[\text{Ag}_{23}\text{Pd}_2(\text{PPh}_3)_{10}\text{Br}_7]^0$, the effects of central atom type and bridging halogen on the electronic structures of the resulting superatomic molecules are discussed. The effects of halogen atoms and bridging ligands on the formation of stable supramolecular molecules are also demonstrated. The manuscript is well written and it is suggested to be revised for publication. There are some suggestions.

Reply 1:

We wish to thank the reviewer for providing valuable comments and suggestions. We have revised the manuscript in accordance with your comments. All the revised portions appear as red text in the revised manuscript.

Comment 2:

Although the UV-vis spectra and XPS spectra of the $[\text{Ag}_{23}\text{Pt}_2(\text{PPh}_3)_{10}\text{Br}_7]^0$ and $[\text{Ag}_{23}\text{Pd}_2(\text{PPh}_3)_{10}\text{Br}_7]^0$ nanoclusters were performed, more characterizations such as ESI, TGA, etc. are desirable to demonstrate the purity of the nanoclusters for the experiments.

Reply 2:

$[\text{Ag}_{23}\text{Pt}_2(\text{PPh}_3)_{10}\text{Br}_7]^0$ and $[\text{Ag}_{23}\text{Pd}_2(\text{PPh}_3)_{10}\text{Br}_7]^0$ are not so stable in dichloromethane solution, and thereby we could not measure ESI-MS spectra for these clusters at the submission stage. However, we found that the stability of 1–4 can be improved when toluene was used as a solvent (Figure 7 in the revised manuscript).

Fig. 7: Time dependence of the optical absorption spectra. a–d 1, 2, 3, and 4, respectively. e Time dependence of plots of absorbance of the first and second peaks in the optical absorption spectra of **1** (446, 560 nm), **2** (489, 630 nm), **3** (452, 560 nm) and **4** (497, 630 nm).

The stability of 1–4 can be further improved when extra PPh_3 was added to the toluene solution of 1–4 (Figure S24 in the revised manuscript). We have described these facts in the main manuscript (line 256–258 in the revised manuscript).

Fig. 24: Optical absorption spectra of cluster toluene solution prepared by dissolving crystals of 1–4 before and after leaving for 1 week. a 1, b 2, c 3 and d 4. In these experiments, an excess amount of PPh₃ (95 mM) was dissolved to suppress the degradation of 1–4 in solution. These spectra demonstrate that the stability of 1–4 in solution is largely improved by the addition of an excess amount of PPh₃ to the solution.

In this way, since the stability of [Ag₂₃Pt₂(PPh₃)₁₀Br₇]⁰ and [Ag₂₃Pd₂(PPh₃)₁₀Br₇]⁰ in the solution was improved, we could obtain ESI-MS spectrum for [Ag₂₃Pt₂(PPh₃)₁₀Br₇]⁰ (Figure S2 in the revised manuscript). We have described the obtained information in the main manuscript (line 65–67 in the revised manuscript). However, we could not still monitor the parent peak for the most unstable [Ag₂₃Pd₂(PPh₃)₁₀Br₇]⁰ due to the lack in the stability.

Fig. S2: Expanded ESI-MS spectra of 3. The inset shows the comparison between experimental and calculated mass distribution for [Ag₂₃Pt₂(PPh₃)₁₀Br₇Cs₂]²⁺. The peaks *1, *2 and *3 are assigned to [Ag₂₃Pt₂(PPh₃)₉(PPh)Br₅Cl₂Cs₂]²⁺, [Ag₂₃Pt₂(PPh₃)₁₀Br₄Cl₃Cs₂]²⁺ and [Ag₂₃Pt₂(PPh₃)₁₀Br₆ClC₂S₂]²⁺, respectively. Since Cl was not observed in SC-XRD analysis on 3 (Fig. 1c), peaks *1, *2 and *3 seem to be produced by the contamination of Cl ions during ESI-MS. In this mass spectrum, a peak progression was observed due to the addition of Cs₂CO₃ as a cation source, similar to the case of the literature.

As an additional characterization, we have measured XPS spectrum of [Ag₂₃Pt₂(PPh₃)₁₀Br₇]⁰ to confirm the central atom to be Pt. We have described this fact in the main manuscript (65–67 in the revised manuscript) and added Pt 4f spectrum of [Ag₂₃Pt₂(PPh₃)₁₀Br₇]⁰ in Figure S3 in the revised manuscript. Regarding TGA, unfortunately, we could not obtain the reliable patterns due to the lack of the quantity of the obtained crystals of [Ag₂₃Pt₂(PPh₃)₁₀Br₇]⁰ (3) and [Ag₂₃Pd₂(PPh₃)₁₀Br₇]⁰ (4).

Fig. S3: Pt 4f spectra of 3. This spectrum indicate that Pt atom is certainly included in 3.

Comment 3:

Whether the structure distortion caused by halogen atoms will affect the fluorescence intensity and emission location?

Reply 3:

We have newly conducted the photoluminescence (PL) measurements for the cluster solution at 25 °C. The results demonstrated that 1) all clusters 1–4 exhibit the PL in visible region and 2) the PL peak positions of the clusters 2 and 4 are red-shifted compared to those of the clusters 1 and 3. This trend is well consistent with that of optical absorption and HOMO-LUMO gap estimated by DFT calculation. We have described this fact in the main manuscript (line 50–51, 177–179 in the revised manuscript) and showed the PL spectra in Figure 6 (in the revised manuscript).

Fig. 6: PL spectra obtained for the cluster solution of 1–4 at 25 °C. The cluster solutions of 1–4 were excited by the light of 451, 487, 459, and 496 nm, respectively. In this figure, the vertical axis is normalized to eliminate the effect of the difference of the concentration of the cluster solution on the PL intensity.

In addition, we have calculated the optical absorption spectrum also for $[Ag_{23}Pt_2(PPh_3)_{10}Br_7]^0$ without distortion and $[Ag_{23}Pd_2(PPh_3)_{10}Br_7]^0$ without distortion (Figure S20 and S21 in the revised manuscript). The results demonstrate that the optical absorption spectrum changes only a little depending on the distortion. We have described the fact in the main manuscript (line 164–168 in the revised manuscript).

Fig. S20: Comparison of the calculated optical absorption spectrum between a $[\text{Ag}_{23}\text{Pt}_2(\text{PPh}_3)_{10}\text{Br}_7]^0$ without distortion and b $[\text{Ag}_{23}\text{Pt}_2(\text{PPh}_3)_{10}\text{Br}_7]^0$ with distortion (3'). In the calculation of $[\text{Ag}_{23}\text{Pt}_2(\text{PPh}_3)_{10}\text{Br}_7]^0$ without distortion, Cl of 1' was replaced with Br.

Fig. S21: Comparison of the calculated optical absorption spectrum between a $[\text{Ag}_{23}\text{Pd}_2(\text{PPh}_3)_{10}\text{Br}_7]^0$ without distortion and b $[\text{Ag}_{23}\text{Pd}_2(\text{PPh}_3)_{10}\text{Br}_7]^0$ with distortion (4'). In the calculation of $[\text{Ag}_{23}\text{Pd}_2(\text{PPh}_3)_{10}\text{Br}_7]^0$ without distortion, Cl of 2' was replaced with Br.

On the basis of these results, we can consider that the PL peak shift is also caused by not the distortion but the kinds of the central atom. We have described this fact in the main manuscript (line 177–179 in the revised manuscript).

Comment 4:

Please address or explain the Alert level A/B.

Reply 4:

We have conducted the re-analyses and thereby the analyses were a little improved. However, there are still Alert level A/B. For these Alerts, we have described the reason why the Alert appears in check-cif (Figure CL2 and CL3). Accordingly, we have replaced cif file and checkcif.

Figure CL2. Renewed check-cif for cluster 3.

Figure CL3. Renewed check-cif for cluster 4.

Comment 5:

“For 1–4, it is difficult to estimate the HOMO–LUMO gap of each superatomic molecule from its optical absorption spectrum because the HOMO–LUMO transition is forbidden (Fig. S14)” (Line 155-156). Please give a more detailed explanation and add references.

Reply 5:

The overlap of the wave function between HOMO and LUMO is zero, therefore, transition dipole moment from HOMO to LUMO becomes zero. This is the reason why the HOMO–LUMO transition is forbidden. We have described this fact in the caption of Figure 5 in the revised manuscript.

Comment 6:

Whether the HOMO-LUMO can be observed from the electrochemical spectra?

Reply 6:

Although we have attempted to experimentally estimate HOMO–LUMO gap by electrochemical measurements, we could not observe a reliable voltammogram due to the lack of the quantity of the obtained crystals. We have described this fact in the main manuscript (line 173–175 in the revised manuscript).

Reviewer: 3

Comment 1:

Miyajima et al. reported the synthesis of two $\text{Ag}_{23}\text{Pt}_2(\text{PPh}_3)_{10}\text{Br}_7/\text{Ag}_{23}\text{Pd}_2(\text{PPh}_3)_{10}\text{Br}_7$ clusters, which own two icosahedral core. These two cores share a common metal atom and are bridged by halogen ligands. The authors then tried to clarify the effects of the central atom and the type of bridging halogen. Although this work was well organized, I cannot recommend the publication of this paper on communication chemistry because the existed limitations affects the rationality of this work. The reasons are given below.

Reply 1:

We wish to thank the reviewer for providing valuable comments and suggestions. We have revised the manuscript in accordance with your comments. All the revised portions appear as red text in the revised manuscript.

Comment 2:

This paper lacks the novelty to publish in a Nature Communication sister journal. As indicated in Table 1, several clusters of the similar structures have been synthesized before. The replacement of Br into Cl does not give us more understanding/guidance for future study.

Reply 2:

To construct substances using superatoms and create new materials, **it is essential to gain a deeper understanding of the types of connected clusters that can be produced and the electronic structures that can be created.** However, in the previous studies, it has not been well understood about 1) which kinds of bi-icosahedral metal cluster can be produced and 2) what is the key point for connecting metal clusters. The synthesis of new bi-icosahedral metal clusters itself is not so great finding. However, we could reveal 1) and 2) by newly synthesizing $[\text{Ag}_{23}\text{Pt}_2(\text{PPh}_3)_{10}\text{Br}_7]^0$ and $[\text{Ag}_{23}\text{Pd}_2(\text{PPh}_3)_{10}\text{Br}_7]^0$ and comparing the stability of four kinds of clusters **1–4**. The obtained understanding well explains the reason why the reported bi-icosahedral metal clusters could be isolated, indicating that the key factor reported in this article is applicable not only to **1–4** but also to the other bi-icosahedral metal clusters. The elucidation of key factor leads to the prediction of the bi-icosahedral metal clusters which are expected to be isolated in the future, such as $[\text{Ag}_{23}\text{PtPd}(\text{PPh}_3)_{10}\text{X}_7]^0$, $[\text{Ag}_{23}\text{Ni}_2(\text{PPh}_3)_{10}\text{X}_7]^0$, $[\text{Ag}_{23}\text{PtNi}(\text{PPh}_3)_{10}\text{X}_7]^0$, and $[\text{Ag}_{23}\text{PdNi}(\text{PPh}_3)_{10}\text{X}_7]^0$ ($\text{X} = \text{Cl}$ or Br) (line **256–258** in the revised manuscript). In this way, **the findings in this work are expected to provide clear design guidelines for the creation of novel materials with various properties and functions in the future. The obtained three information, 1–3, should also be useful for the future work on the isolation of metal clusters composed of three or four superatoms.** Accordingly, we believe that the manuscript merits consideration as an Article in *Communications Chemistry*. Thus, please reconsider your conclusion. We have described the novelty and extendability of this work in the main manuscript (line **261–263** in the revised manuscript).

Comment 3:

The major discovery of this work indicates that halogen ligand with proper size is essential for the formation and isolation of a superatomic molecule consisting of two $\text{Ag}_{13-x}\text{M}_x$ structures ($\text{M} = \text{Ag}$ or other metal). Although not tried in this work, it cannot be totally excluded if thiolate ligand can achieve the same goal because Jin et al. ever prepared $[\text{Ag}_2\text{Au}_{23}(\text{PPh}_3)_{10}(\text{PET})_5\text{Cl}_2]^{2+}$ with the utilization of thiolate ligand. Thus, halogen ligand can be one of the practicable ways but not essential.

Reply 3:

We also agree with the reviewer. Therefore, we are describing the fact that even more stable superatomic molecules can be obtained if SR and SeR are used as bridging ligands for superatomic molecules composed of $\text{Au}_{13-x}\text{M}_x$ in the main manuscript (line **269–273** in the revised manuscript).

Comment 4:

The authors state icosahedral core with heteroatom substitution is essential. However, to my understanding, the pure Ag cluster is still await to be prepared. I understand icosahedral core with heteroatom substitution can be more stable but this does not mean pure Ag cluster is impossible to be obtained. Because Bakr et al. ever obtained $[\text{Ag}_{25}(\text{SR})_{18}]^-$, which owns icosahedral core.

Reply 4:

Regarding $[\text{Ag}_{25}(\text{PPh}_3)_{10}\text{Cl}_7]^{2+}$ and $[\text{Ag}_{25}(\text{PPh}_3)_{10}\text{Br}_7]^{2+}$, we have again attempted to synthesize those which changing the experimental conditions. The results demonstrated that somethings were synthesized just after adding NaBH_4 even without adding the precursor salts of heteroatoms (H_2PtCl_6 , $\text{Pd}(\text{PPh}_3)\text{Cl}_6$, PtBr_2 or PdBr_2), since the solution color became yellow. However, the solution became soon colorless and the black precipitate was obtained. According to these results, it can be considered that the stability is quite low even

if the clusters can be formed in solution for $[\text{Ag}_{25}(\text{PPh}_3)_{10}\text{Cl}_7]^{2+}$ and $[\text{Ag}_{25}(\text{PPh}_3)_{10}\text{Br}_7]^{2+}$. Namely, the stability decreases in the order $1 > 3 > 2 > 4$. We have described this fact in the main manuscript (line 193–197 in the revised manuscript). However, we can also understand what you want to say. Therefore, we weakened the description for $[\text{Ag}_{25}(\text{PPh}_3)_{10}\text{Cl}_7]^{2+}$ and $[\text{Ag}_{25}(\text{PPh}_3)_{10}\text{Br}_7]^{2+}$ from “which are difficult to synthesise” to “which are not so stable in solution” (line 53–54, 200–201 in the revised manuscript).

Comment 5:

A key flaw of this work. The author suggest the stability increases in the order $1 > 3 > 2 > 4$. However, it is hard to obtain this conclusion because the time dependence of the optical absorption spectra of 3 in Figure 5 shows the least variation to me. This may affect the discussions because substantial efforts were paid to rationalize the effects of different halogen ligands.

Reply 5:

Regarding the solution stability, we conducted additional measurements using toluene instead dichloromethane as a solvent. The results improved the stability of each cluster 1–4, clearly demonstrated that the stability decreases in the order $1 > 3 > 2 > 4$. Accordingly, we have renewed Figure 5 (Figure 7 in the revised manuscript) and furthermore made the statistic panels for each cluster 1–4, which clearly demonstrate the stability order, and added these statistic panels in Figure 7e. We have described the obtained results in the main manuscript (line 184–192 in the revised manuscript).

Fig. 7: Time dependence of the optical absorption spectra. a–d 1, 2, 3, and 4, respectively. e Time dependence of plots of absorbance of the first and second peaks in the optical absorption spectra of **1** (446, 560 nm), **2** (489, 630 nm), **3** (452, 560 nm) and **4** (497, 630 nm).

Comment 6:

Several supporting figures (even in the first paragraph) were used to show the background of this work. To my understanding, this is not common. Instead, the authors can try to give citations in the introduction.

Reply 6:

We have modified Introduction. Accordingly, all the supporting figures used for the background of this work were removed.

Comment 7:

1', 2', 3', and 4' can be removed because they indicate the same cluster with 1, 2, 3, and 4. p.w. can be used to indicate present work instead of '-'.

Reply 7:

We have modified Table 1.

Table 1 NC number, chemical composition, number of bridging halogen, number of total valence electrons and literature on silver-based di-superatomic molecule described in this paper

NC	Chemical composition ^a	N_{bx} ^b	N_{te} ^c	Ref.
1	[Ag ₂₃ Pt ₂ (PPh ₃) ₁₀ Cl ₇] ⁰	5	16	25
2	[Ag ₂₃ Pd ₂ (PPh ₃) ₁₀ Cl ₇] ⁰	5	16	24
3	[Ag ₂₃ Pt ₂ (PPh ₃) ₁₀ Br ₇] ⁰	5	16	p.w. ^d
4	[Ag ₂₃ Pd ₂ (PPh ₃) ₁₀ Br ₇] ⁰	5	16	p.w. ^d
5	[Au ₂₃ Pd ₂ (PPh ₃) ₁₀ Br ₇] ⁰	5	16	36
6	[Au ₁₃ Ag ₁₂ (P(p -Tol) ₃) ₁₀ Cl ₇](SbF ₆) ₂	5	16	37
7	[Au ₁₃ Ag ₁₂ (PPh ₃) ₁₀ Cl ₈](SbF ₆)	6	16	38
8	[Au ₁₃ Ag ₁₂ (PPh ₃) ₁₀ Br ₈](SbF ₆)	6	16	39
9	[Au ₁₃ Ag ₁₂ (P(p -Tol) ₃) ₁₀ Br ₈](PF ₆)	6	16	40
10	[Au ₁₃ Ag ₁₂ (PPh ₃) ₁₀ Br ₈]Br	6	16	41
11	[Au ₁₃ Ag ₁₂ (PMePh ₂) ₁₀ Br ₉] ⁰	7	16	42
12	[Au ₁₁ Ag ₁₂ Pt ₂ (PPh ₃) ₁₀ Cl ₇] ⁰	5	16	45
13	[Au ₁₀ Ag ₁₃ Pt ₂ (PPh ₃) ₁₀ Cl ₇] ⁰	5	16	46

^a For example, [Ag₂₃Pt₂(PPh₃)₁₀Cl₇]⁰ (calc.) indicates the DFT-calculated [Ag₂₃Pt₂(PPh₃)₁₀Cl₇]⁰. ^b Number of bridging halogens. ^c Number of total valence electrons. ^d Present work

Comment 8:

The authors used the “ionic radius”, but the cited reference actually means “atomic radius”.

Reply 8:

Although the title of the literature includes “atomic radius”, we used the value of “ionic radius” reported in this literature.

Comment 9:

How did the authors get the conclusion of “L4 and L6 layers do not own strong correlation”. The corresponding Ag-Ag distance should be within 3.0 Å.

Reply 9:

We mean that the results showed **no strong correlation between the charge repulsion in the L4–L6 layer and the stability of 1–4 and [Ag₂₅(PPh₃)₁₀X₇]²⁺ (X = Cl or Br)**. This is because 1) the magnitude of charge repulsion was estimated to be in the order 1' > 5' > 2' = 3' > 4' = 6' and 2) this order does not consistent with that of the stability (1 > 3 > 2 > 4). We have describe that fact in the main manuscript (line 217 in the revised manuscript).

REVIEWERS' COMMENTS:

Reviewer #2 (Remarks to the Author):

The authors answered the reviewers' comments and revised the manuscript properly. The quality of manuscripts has really improved. I think the manuscript is acceptable as it.

Reviewer #3 (Remarks to the Author):

The authors well addressed my major concerns. Current version can be acceptable for publication as is.